# Bayesian Analysis of the Association between Casein Complex Haplotype Variants and Milk Yield, Composition, and Curve Shape Parameters in Murciano-Granadina Goats

**DOI:** 10.3390/ani10101845

**Published:** 2020-10-10

**Authors:** María Gabriela Pizarro Inostroza, Francisco Javier Navas González, Vincenzo Landi, Jose Manuel León Jurado, Juan Vicente Delgado Bermejo, Javier Fernández Álvarez, María del Amparo Martínez Martínez

**Affiliations:** 1Department of Genetics, Faculty of Veterinary Sciences, University of Córdoba, 14071 Córdoba, Spain; z12piinm@uco.es (M.G.P.I.); id1debej@uco.es (J.V.D.B.); ib2mamaa@uco.es (M.d.A.M.M.); 2Animal Breeding Consulting, S.L., Córdoba Science and Technology Park Rabanales 21, 14071 Córdoba, Spain; 3Department of Veterinary Medicine, University of Bari “Aldo Moro”, 70010 Valenzano, Italy; vincenzo.landi@uniba.it; 4Centro Agropecuario Provincial de Córdoba, Diputación Provincial de Córdoba, Córdoba, 14071 Córdoba, Spain; jomalejur@yahoo.es; 5National Association of Breeders of Murciano-Granadina Goat Breed, Fuente Vaqueros, 18340 Granada, Spain; j.fernandez@caprigran.com

**Keywords:** αS1-casein, αS2-casein, β-casein, κ-casein, protein, fat, dry matter, lactose, somatic cells count

## Abstract

**Simple Summary:**

Although the casein complex has been reported to condition economically important traits in dairy species, information in goats is scarce. The analysis of haplotypic sequences has been suggested to maximize the results obtained after the assessment of other genetic units, such as SNPs. We studied the association between haplotype sequences for αS1-, αS2-, β-, and κ-casein loci and milk yield, protein, fat, dray matter, lactose, somatic cells count, and the curve shape parameters that they describe during lactation (peak and persistence). This aimed to provide basic research data for the integration of haplotype monitoring in the selection strategies of dairy goats.

**Abstract:**

Considering casein haplotype variants rather than SNPs may maximize the understanding of heritable mechanisms and their implication on the expression of functional traits related to milk production. Effects of casein complex haplotypes on milk yield, milk composition, and curve shape parameters were used using a Bayesian inference for ANOVA. We identified 48 single nucleotide polymorphisms (SNPs) present in the casein complex of 159 unrelated individuals of diverse ancestry, which were organized into 86 haplotypes. The Ali and Schaeffer model was chosen as the best fitting model for milk yield (Kg), protein, fat, dry matter, and lactose (%), while parabolic yield-density was chosen as the best fitting model for somatic cells count (SCC × 103 sc/mL). Peak and persistence for all traits were computed respectively. Statistically significant differences (*p* < 0.05) were found for milk yield and components. However, no significant difference was found for any curve shape parameter except for protein percentage peak. Those haplotypes for which higher milk yields were reported were the ones that had higher percentages for protein, fat, dry matter, and lactose, while the opposite trend was described by somatic cells counts. Conclusively, casein complex haplotypes can be considered in selection strategies for economically important traits in dairy goats.

## 1. Introduction

The inclusion of new technologic advances in breeding programs gradually evolved from the implementation of traditional phenotypic selection to genomic selection methods through single nucleotide polymorphisms (SNPs). The study of the association of SNPs may allow identification of which gene sequences may associate with goat milk production, quality, and composition, as well as cheesemaking properties [1,2,3]. As a result, geneticists are able to identify and select those individuals with superior genetic potential [1,2].

The genes comprising the casein complex are located within a 250-kb segment on chromosome 6 in the goat [4]. Concretely, SNPs have been reported to act as genetic units, which are closely bonded through epistatic relationships [5] and transmitted as haplotypes [6]. It is the relationship among the genetic polymorphisms of the casein complex (α_S1_, β, α_S2_, and κ-casein genes) with the aforementioned characteristics of productive interest that shapes one of the most interesting complexes to study from an economic perspective [7].

Although the close association between casein genes, casein variants, and milk production traits has been frequently considered, the consideration of casein haplotype variants rather than the coding of a single gene or SNPs, as suggested by several authors, may maximize the understanding of heritable mechanisms and their implication on the expression of functional traits related to milk production [8,9].

The selection of one desirable allele over the rest normally implies the simultaneous selection and frequency increase of epistatically linked alleles, which in turn may not have a favorable effect on milk production traits and their components [10]. The approaches that are regularly followed to perform association analyses [11] normally overlook this genetic epistatic effects change during lactation [12], which renders the study of genetic association inefficient.

Furthermore, previous studies have suggested the expression of casein genes may change across the different phases of lactation in dairy animals [13,14]. This finding potentially implies the fact that the epistatic relationships between the polymorphisms within the genes comprised in the casein complex may change in time and combinedly promote changes in the expression of milk yield and composition along the lactation [15]. As a result, when the regulation of the expression of a certain casein gene has a detrimental effect on milk yield and quality, the remaining genes may positively adjust to compensate for such a negative effect [16].

Such evolution across lactation and the strong dependence of qualitative and quantitative dairy performance on the shape of the lactation curve [11] and its parameters (slope of the initial rise of the curve, peak yield, time to peak, lactation persistency, and length as suggested by López et al. [17]) of the lactation curve shape may be potential selection criteria to consider when the aim is to maximize the profitability of goat milk performance.

The wide interindividual variation in terms of the lactation curve shape (peak yield and persistency) described by individuals may be conditioned by a multifactorial genetic and nongenetic basis [11]. Concretely, the genetically conditioned fraction of these traits [18] may be supported by the findings by Strucken et al. [15], who reported that lactation curve parameters provide a higher power to screen the whole genome for the region whose effect changes during lactation.

Revealing which genetic units may be responsible for the higher productive success though the modelling of the lactation curve may translate in a higher final profitability of milk as a product.

Although several linear and non-linear functions have been used to describe the relationship between daily milk yield and days in milk (DIM) in dairy animals [19], Ali–Schaeffer polynomials may be preferable in genotyping and genetic evaluation studies [20] aiming at modelling for milk yield and composition, while parabolic functions have been suggested to overcome other models when somatic cell count cycles are modelled [21]. These models have been reported to overcome the remaining possibilities when researchers are compelled to use reduced samples, or which require the modelling of individual lactation curves, provided their greater ability to capture inter and intra individual variability and predictability.

Therefore, the objective of this study was to identify casein complex haplotype variants and their repercussions on milk yield, composition, and the parameters of the lactation curve model for milk yield, protein, fat, dry matter, lactose, and somatic cell count in Murciano-Granadina goats.

## 2. Materials and Methods

### 2.1. Animal Sample and Sample Selection Process

A total of 159 herdbook-registered (Delgado et al., 2005) Murciano-Granadina goats were considered in the analysis. Animals in the sample belonged to 28 farms in the south of Spain, whose records were collected in random periods, from October 2006 to June 2018. The mean age of the animals in the sample was 1.39 years old (from 1 year to 9.15 years).

### 2.2. Milk Yield Standardization and Composition Analysis

The Murciano-Granadina polyestric reproductive annual cycle features two kidding seasons, with lactations that frequently last longer than 210–240 days. Total milk yield was standardized to 210 days in milk (DIM) and expressed in kg as described in Pizarro et al. [22] following the premises described in the statutes of CAPRIGRAN (National Association of Breeders of Murciano-Granadina Goat Breed), given the methods performed have proven to be as accurate as the Fleischmann method as required in the guidelines in ICAR [23].

Real milk production (RP_j_) for each goat was calculated using the function and methods described in Pizarro Inostroza et al. [3]. Royal Decree-Law 368/2005 of the Spanish Ministry of Agriculture set the regulations for official milk controls for the genetic evaluation in bovine, ovine, and caprine species on 8 April 2005. Milking took place every 4 or 6 weeks, either in the morning or in the afternoon. For each individual considered in the analyses, the first control (d_1_) and the last (d_2_) were calculated by computing the days between the day the animal gave birth (KD) and the date of the first control (FC), and the days between the control before the last control (PC) and the last one (LC), as described in Pizarro Inostroza et al. [3]. Official monthly sampling was performed at the Milk Quality Laboratory, in Cordoba (Spain) to quantify protein, fat, solids, and lactose content with a MilkoScan™ analyzer FT1, while a somatic cell count (SCC) was performed using Fossomatic™ FC. Milk components were expressed as % while SCC were expressed as SCC × 10^3^ sc/mL.

### 2.3. Milk Production Records

A mean (±SD) of 3.91 ± 2.01 lactations per goat were considered (3107 milking records, 399 lactations) in the statistical evaluation. A mean of 21 days from kidding to first control were computed. The mean number of controls per lactation were 5. Normality was tested using the Shapiro–Francia test with the Stata Version 15.0 software, while Levene’s test for variance homogeny of the SPSS Statistics for Windows statistical program, Version 25.0 was used to test for homoscedasticity.

### 2.4. Selection of Best-Fitting Milk Yield and Composition Curve Models and Curve Shape Parameters Calculation

Forty-nine linear and non-linear models were tested to determine the best-fitting model for individual lactation curves for milk yield and components (Appendix A). The non-linear regression task from the regression procedure of SPSS version 25.0 [24] was used to determine the fitting properties of each model. A Levenberg–Marquardt iterative process using the Regression procedure of SPSS version 25.0 (Corp., 2017) was used to perform an initial search grid to cover curve the shape parameter bounds of each model (b0, b1, b2, b3, and b4 parameters). The rounds of iteration continued until a tolerance convergence criterion of 10^−8^ was reached as suggested in the literature [25]. Once the convergence criterion was determined, initial parameters were set, and model fitting properties were evaluated. The maximum number of iteration rounds used for each analysis was 2000 as suggested in IBM SPSS Statistics Algorithms version 25.0 by IBM Corp. [26]. The mean round number to reach convergence was 3.158 ± 0.682 (μ ± SD).

To determine which function or functions should be used to model milk yield and components out of the 49 models tested, residual values were computed after the result from the difference between the observed value and predicted values. The normality of residuals was tested using the Shapiro–Francia test. The Durbin–Watson test [27] was conducted on the residuals of each model (using average yields of each day of lactation) to test for potential first-order autocorrelations among residuals. The Linear regression test of the regression procedure in SPSS version 25.0. provided the Durbin–Watson statistic. Among the criteria considered to choose the best-fitting model, the percentage of successfully fitted curves (100%); residual sum of squares (RSS), which measures the error that differs from the regression function to the data set that was tested; mean square prediction error, which measures error variation provided its better suitability that other affine parameters in cases of reduced sample sizes [28]; and adjusted R squared or modified R squared (Adj. R^2^), which measures the predictive ability for new observations. Additionally, the Akaike information criterion (AIC), corrected Akaike information criterion (AICc), and Bayesian information criterion (BIC) were compared across models to determine the model presenting the greater ability to explain or capture the variability observed in the data set being studied (AIC and AICc) and the predictive potential (BIC) of each model, respectively. Once the best-fitting models were selected, peak and persistency were calculated. Peak yield and persistency for milk yield and composition were computed as reported in Table 1. The Ali and Schaeffer model was chosen as the best-fitting model for milk yield (kg), protein, fat, dry matter, and lactose (%) and parabolic yield-density was chosen as the best fitting model for somatic cell count (SCC × 10^3^ sc/mL).

### 2.5. Genotyping and Linkage Disequilibrium (LD)

A modification of the procedure described in Miller et al. [31] was performed for DNA isolation. To this aim, 16 nonrelated does were randomly chosen from the herdbook of the breed. Oligonucleotide sequences and SNPs promoters, UTRH3′ regions, and polymorphic exons are described in Pizarro Inostroza et al. [3]. A Platinum High Fidelity (LifeTechnology, Carlsbad, CA, USA) PCR kit was used to amplify polymorphic regions. MACROGEN sequencing service (Macrogen Inc., Korea) sequenced the PCR product and MEGA7 software and Ensembl Genome Browser 97 database were used to analyze pherograms and evaluate previous annotations for SNPs [32]. Genotyping was performed using the Kompetitive allele specific PCR (KASP) assay (LGC Limited, Fordham, UK) and KlusterCaller software (LGC Limited, Fordham, UK). Heterozygosity values of around 40% suggested the number of SNPs to be used as genomic controls was enough [33] so as to prevent the effects from population stratification.

Minor allele frequency (MAF) was calculated to differentiate between common and rare variants (MAF < 0.05) using PLINK v1.90 [34]. Casein complex SNPs’ linkage disequilibrium extent (LD) was calculated using HaploView software (Broad Institute^TM^, Cambridge, MA, USA) [19], scoring LD through D’ (normalized linkage disequilibrium coefficient) and r^2^ (linkage disequilibrium coefficient of determination) (Appendix A). The total length of casein loci and distances between adjapcent loci were determined following the premises presented by Dagnachew et al. [35].

### 2.6. Haplotyping

Phasing (or haplotyping) describes the process of determining haplotypes from the genotype data [36]. As suggested by Glusman et al. [36], haplotypes are more specific than less complex variants, such as single nucleotide variants (SNP variants). A haplotype-based empirical model inherited from an SNP-based method was followed as suggested by Chen et al. [37]. We identified 48 single nucleotide polymorphisms (SNPs) present in the casein complex of 159 unrelated individuals of diverse ancestry, which were organized the SNPs into 86 different haplotypes. Haplotype sequences can be consulted in Appendix A. The results from the analyses of epistatic relationships in Pizarro Inostroza et al. [5] were also considered for validation of those that were determined.

### 2.7. Bayesian Analysis of Haplotype Factor

Although the simplest and perhaps the most frequently used test for parametric associations considers the relationship between a single marker and a quantitative trait, the power of this method may suffer because a single SNP may have only low linkage disequilibrium (LD) with the causal mutation, and the LD contained jointly in flanking markers is ignored. Alternatively, fitting hereditary units, such as SNPs or haplotypes, simultaneously using Bayesian methods, thus considering the LD between neighboring SNPs, may prevent a false positive from occurring [38].

In this context, Bayesian inference for ANOVA was run to test for statistical differences in the mean across haplotype variants (independent factor) for milk yield and composition and the curve shape parameters (dependent variables) provided the limitations in sample size and given parametric assumptions were violated (*p* < 0.05).

No interaction between haplotypic variants was considered as suggested by papers studying independent factors and the effects of their double and triple interactions. For instance, the consideration of factors independently may help quantify their effects more accurately than their conjoint effects [39]. This may be ascribed to the fact that when a limited sample is available, some of the potential combinations across categorical factors may be misrepresented [40].

According to Cleophas and Zwinderman [41], in traditional ANOVA, the mean values per group are squared and after adjustment for degrees of freedom divided by the squared standard deviations of the groups. The F-statistic should be larger than approximately 5 for the null hypothesis of no difference between variability of the treatment groups to be rejected, as tested against the variability of the subjects (within the groups). Contrastingly, Bayesian ANOVA assesses the magnitude of the Bayes factor (BF) as computed from the ratio of a posterior and prior likelihood distribution. The posterior is modelled from the means and variances of the measured unpaired groups, the prior is usually modelled as an uninformative prior either from the Jeffreys–Zellener–Siow (JZS) method, or, equivalently, from the computation of a reference prior based on a gamma distribution with a standard error of 1. The computation of the BF requires integral calculations for accuracy purposes. However, then, it can be used as a statistical index that pretty precisely quantifies the amount of support in favor of either H_1_ (the difference between the unpaired means is larger than zero) or H_0_ (the difference between the unpaired means is not larger than zero). Among the advantages of the Bayesian approach, it provides a better underlying structure model of the H_1_ and H_0_ and the maximal likelihoods of likelihood distributions are not always identical to the mean effect of traditional tests, and this may be fine, because biological likelihoods may better fit biological questions than numerical means of non-representative subgroups do.

As suggested in the public document IBM SPSS Statistics Algorithms version 25.0 by IBM Corp. [26], Bayesian inference of ANOVA is approached as a special case of the Bayesian general multiple linear regression model. A full description of the algorithms used by SPSS to perform Bayesian Inference on Analysis of Variance (ANOVA) in this study can be found in the public document IBM SPSS Statistics Algorithms version 25.0 by IBM Corp. [26]. The tolerance value for the numerical methods and the number of method iterations were set as a default by SPSS v25.0 (IBM Corp., Armonk, NY, USA) [24].

Posterior distribution statistics and their 95% credibility intervals (CI) were computed to determine the magnitude of the effect of the haplotype factor on milk yield and composition and to identify those haplotypes and which may be linked to a superior performance. A significant effect was denoted when 0 was not comprised within CI.

The Bayes factor (BF) measures the likelihood of null and alternative hypotheses (strength of the evidence) and is used instead of *p* values (from frequentist approaches) to draw conclusions. Still, as suggested by Cleophas and Zwinderman [41], extrapolation between the Bayes factor used in Bayesian approaches and *p* values from frequentist approaches could be performed to favor the interpretability of results. The larger the BF, the more the evidence favors the alternative hypothesis compared to the null hypothesis.

The JZS prior was used as it is appropriate for Bayesian inference of ANOVA, provided its characteristic symmetry is around zero [42]. As a result, positive and negative values of the slope parameters a priori have the same probability of occurring. Additionally, this prior scale invariant, hence deriving the Bayes factor, is not scale dependent.

## 3. Results

In the context of the scarcity of studies evaluating haplotypes in goats for the traits evaluated in the present paper, the analysis of results was performed through the comparison of haplotypic sequences. Haplotypic sequences were compared to identify which sections within them were common or, on the contrary, which varied across haplotypic variants. As reported in Appendix A, a color code was assigned at random to identify the same haplotypic sequence across haplotypic variants. Afterwards, significant differences in the levels of traits (milk yield and composition and their curve shape parameters) were ascribed to the differences found in the haplotypic sequences (allelic variants) across haplotypic variants. Provided the fact that the lower number of haplotypic variants was found for αS1-casein, the results will be presented accordingly, comparing this to all potential haplotypic combinations found across the rest of casein complex loci.

### 3.1. Milk Yield and Composition Association with Potential Combinations of αS1- and αS2-Casein Loci Haplotypic Sequences

Table 2 shows results of Bayesian inference of ANOVA to detect significant difference in average milk yield, protein, fat, dry matter, lactose, and somatic cell counts and curve shape parameters for milk yield and each of the aforementioned components across casein complex haplotypes. Statistically significant (*p* < 0.05) differences were found for milk yield and all components, but among the curve shape parameters (peaks and persistence), only average protein percentage peak reported significant differences across casein complex haplotypes. Appendix A shows the descriptive statistics for milk yield and components across all the haplotypes detected within the casein complex in Murciano-Granadina goats. When the haplotypic sequence found at the αS1-casein locus is considered, four different haplotypic groups can be determined. A map of the linkage disequilibrium relationships across casein complex loci is shown in Figure 1.

First, a significantly higher average milk yield of 3.38 kg was found for haplotypes presenting the sequence AAGGAATTAAAAGGCCAA at the αS1-casein locus, when compared to those presenting GAGAAATCGAGAAAGCAA (2.61 kg), GAGAAATCGAGAGAGCGA (2.06 kg), and GAGGAATTAAAAGAGCAA (2.41 kg). The respective average fat percentage found for such sequences was 5.51%, 5.41%, 5.44%, and 5.16%. The average protein percentage for the same respective sequences was 3.56%, 3.53%, 3.46%, and 3.44%. The average dry matter percentage for the same sequences was 14.70%, 14.68%, 14.55%, and 14.19%, respectively. Lactose described the same trend, with the average lactose percentages being 4.97%, 4.84%, 4.83%, and 4.79%, respectively. Somatic cells counts were respectively 513.80 × 10^3^, 1012.63 × 10^3^, 1130.56 × 10^3^, and 1139.81 × 10^3^ sc/mL. These sequences differed in the bases in positions 1, 4, 15, and 16 within the allelic sequence, which correspond to alleles A→G, G→A, C→G, and C→G in SNPs 19, 26, 28, and 29, respectively (→ indicates the changes in allelic sequences).

Second, for those alleles that presented the common sequence of GAGGAATTAAAAGAGCAA for αS1-casein, the highest average milk yield (2.86 kg) was reported when the αs2-casein locus carried the haplotypic sequence TCGCGGCCAAGACCGAGG, followed by those carrying the sequence TCGCGATCGAGACCGAGC (2.68 kg) and CCGGGGCCAAGGCCAAGG (1.88 kg). The average protein percentage for these sequences was 3.65%, 3.56%, and 3.51%, respectively, while the average percentage of dry matter for the same sequences was 14.72%, 14.52%, and 14.30%. The average lactose percentages found were respectively 4.90%, 4.79%, and 4.78%. These sequences differed in the bases in positions 1, 4, 13, and 15, within the allelic sequence, which correspond to alleles T→C, C→G, A→G, and G→A in SNPs 2, 3, 10, and 12, respectively (→ indicates the changes in allelic sequences). Milk yield decreased by almost around 1 kg when any of the alleles changed to G or A.

Third, for the haplotypes carrying the GAGAAATCGAGAGAGCGA sequence at the αS1-casein locus, milk yield differed depending on the sequence present in αS2-casein. Haplotypes presenting the TCGCGGCCAAGGCCAAGG sequence at the αS2-casein locus reported an average milk yield of 2.40 kg and progressively decreased when the sequence changed to TTCCGATCGAGACCGACC (2.39 kg), TTCCAATTGGGACCGGCC (1.89 kg), or TTCCGATCGAGACCGGCC (1.47 kg). The results for fat reported the inverse situation, with the aforementioned sequences being associated with average percentages of 5.07%, 5.14%, 5.64%, and 5.88%, respectively, while for average protein percentages, these were 3.99%, 3.57%, 3.53%, and 3.42%, respectively. Average dry matter percentages were respectively 15.43%, 14.83%, 14.25%, and 14.17%. Lactose average percentages were TTCCGATCGAGACCGACC (4.97%), TTCCGATCGAGACCGGCC (4.83%), TTCCAATTGGGACCGGCC (4.79%), and TCGCGGCCAAGGCCAAGG (4.75%), respectively. Somatic cell counts for the same sequences were as follows TTCCGGCCAAGACCGAGG (456.83 × 10^3^ sc/mL), TCGCGGCCAAGGCCAAGG (506.16 × 10^3^ sc/mL), CCGGGGCCAAGGCCAAGG (847.15 × 10^3^ sc/mL), and TTCCGATCGAGACCGGCC (1860.49 × 10^3^ sc/mL). The progressively higher milk yields occurred when C, G, G, C, A, G, and A were present in SNPs 2, 3, 4, 5, 6, 10, and 12 and the GG polymorphism at SNP 17.

Fourth, when the haplotypic sequence GAGGAATTAAAAGAGCAA at the αS1-casein locus was followed by the sequence TTCCGGCCAAGACCGAGG (3.40 kg) of the αS2-casein locus, a significantly higher milk yield was found when compared to other sequences TTCCGATCGAGACCGGCC (1.64 kg), TCGCGGCCAAGGCCAAGG (2.17 kg), and CCGGGGCCAAGGCCAAGG (2.43 kg), respectively. The average fat percentage for the aforementioned sequences was 5.51%, 5.42%, 5.36%, and 5.29, respectively, with the sequence reporting the lowest average percentage of fat simultaneously being the one reporting the highest milk yield as well (TTCCGGCCAAGACCGAGG). This pattern repeated for the average protein and dry matter percentages as follows: 3.18% protein/14.38% dry matter (TTCCGGCCAAGACCGAGG), 3.42% protein/13.97% dry matter (CCGCGGCCAAGGCCAAGG), 3.45% protein/14.19% dry matter/4.77% lactose/940.29 × 10^3^ sc/mL (CCGGGGCCAAGGCCAAGG), 3.57% protein/13.93% dry matter/1278.90 × 10^3^ sc/mL (TCGCGGCCAAGACCGAGG), 3.63% protein/14.38% dry matter/917.33 × 10^3^ sc/mL (TTCCGATCGAGACCGGCC), and 3.67% protein/14.57% dry matter/4.79% lactose/1981.14 × 10^3^ sc/mL (TCGCGGCCAAGGCCAAGG). The sequences differed as some presented the alleles C, G, G, C, A, G, A, and G at SNPs 2, 3, 4, 5, 6, 10, 12, and 17, respectively.

### 3.2. Milk Yield and Composition Association with Potential Combinations of αS1- and β-Casein Loci Haplotypic Sequences

When the αS1-casein sequence GAGAAATCGAGAAAGCAA was associated to the sequence GGGATCTC of the β-casein locus, a higher milk yield was reported (2.45 kg) in comparison to the sequence GGGACCCC (2.34 kg). For these sequences, the percentage of fat reported was as follows: GGGACCCC (5.48%), GGAACCCC (5.45%), and GGGATCTC (5.29%) while the average percentages of protein were GGGACCCC (3.61%), GGAACCCC (3.56%), and GGGATCTC (3.78%). Average dry matter percentages were GGAACCCC (14.85%), GGGATCTC (14.46%), and GGGACCCC (14.77%), respectively. The sequence GGGACCCC presented an average lactose percentage of 4.88%, contrasting the slightly lower percentage of 4.80% reported for GGGATCTC, while the somatic cell counts were GGGACCCC (760.15 × 10^3^ sc/mL) and GGGATCTC (645.96 × 10^3^ sc/mL). Similarly, when the αS1-casein sequence GAGAAATCGAGAGAGCGA was associated with the β-casein locus sequence GGGATCTC (2.63 kg), there was an increase in milk yield, which, however, reduced the average fat/protein/dry matter percentages when followed by the β-casein locus sequence GGGGCCCC (2.03 kg), which parallelly decreased with the increase in milk yield as follows: GGGATCTC (4.91% fat/3.50% protein/13.95% dry matter/4.86% lactose/1148.89 × 10^3^ sc/mL), GGGGCCCC (5.56% fat/3.97% protein/15.19% dry matter/4.59% lactose/959.09 × 10^3^ sc/mL), and GGAATCTC (5.82% fat/3.63% protein/15.13% dry matter), respectively. The last combination corresponds to the combination of the αS1-casein sequence GAGGAATTAAAAGAGCAA with the β-casein sequences GGGGCCCC for which the milk yield reported was 2.96 kg, GGGACCCC (3.73 kg), GGGATCTC (2.90 kg), and GAGACCCC (3.31 kg). A negative correlation was found between milk yields and fat/protein/dry matter percentages, which parallelly decreased with the increase in milk yield as follows: GGGATCTC (6.21% fat/3.43% protein/14.90% dry matter), GGGACCCC (5.02% fat/3.35% protein/14.02% dry matter/744.33 × 10^3^ sc/mL), GGGGCCCC (5.44 % fat/3.50% protein/13.96% dry matter/1255.74 × 10^3^ sc/mL), and GAGACCCC (5.13% fat/3.52% protein/14.25% dry matter/788.10 × 10^3^ sc/mL), respectively. The sequence GGAATCTC was associated with an increased average lactose percentage of 4.88% in comparison to the rest. The aforementioned sequences differed in the change of the alleles A→G, A→G, T→C and T→C at SNPs 34, 35, 36, and 37.

### 3.3. Milk Yield and Composition Association with Potential Combinations of αS1- and κ-Casein Loci Haplotypic Sequences

A wide variability was found in regards to the clusters of sequences for αS1-casein that were combined with those of κ-casein. Still, when the cluster characterized by the αS1-casein sequence GAGAAATCGAGAAAGCAA was combined with the κ-casein sequence TTCCCCAA.-.-GGTTCC (2.85 kg), milk yield increased significantly. The later sequence presented the alleles C, C, A, .-, G, and T, in contrast to the others, which presented T, T, T, AATC, A, and G at SNPs 39, 40, 41, 42, 43, and 44, respectively. Other haplotypes described the same variability. For instance, for the haplotypes clustered together considering the αS1-casein sequence GAGAAATCGAGAGAGCGA, it was not possible to define a clear trend in the association with milk yield, although statistically significant differences were found. As for the rest of the haplotypic sequences found in other loci of the casein complex, for κ-casein haplotypic sequences TTTTTTTTAATCAATCAAGGCC (6.07% fat/3.87% protein), TGCCTCAA.-.-GGTGTC (5.55% fat/3.33% protein), TTTCTCTA.-AATCGATGCC (5.33% fat/3.60% protein), and TTCCCCAA.-.-GGTTCC (5.12% fat/3.13% protein), the same negative correlation between average milk yield and fat/protein percentage was found.

In this widely variable context, when the sequence for the αS1-casein GAGGAATTAAAAGAGCAA was associated with the κ-casein sequence TTTCTCTA.-AATCGATGCC, milk yield slightly increased (2.49 kg) in comparison to when it was associated with the sequence TGTCTTTA.-AATCGAGGTC (2.20 kg). The sequences differed in regards to the presence of the alleles T, C, and C for SNPs 38, 40, and 44. Parallelly, the κ-casein sequences TTCCCCAA.-.- GGTTCC were associated with average fat contents of 5.39%, TTTCTCTA.-AATCGATGCC (4.91%), and TGTCTTTA.-AATCGAGGTC (5.08%), respectively, when the configuring alleles were instead G, T, and T for SNPs 38, 40, and 44. The great variability found in the case of average protein and dry matter percentages does not allow identification of association trends across haplotypes when κ-casein sequences are considered. Although a wide variability was also found for the average lactose percentage, the κ-casein sequence TTCCCCAA.-.-GGTTCC was associated with a slightly higher average percentage of lactose (4.87%) than that reported for the sequence TTTCTCTA.-AATCGATGCC (4.68%). The two sequences differed in regards to the presence of the alleles C, C A, .-, A, and T at SNPs 39, 40, 41, and 42. The same situation was found for the somatic cell counts, where a decreasing trend in a widely variably context was found. For instance, for the sequence TTCCCCAA.-.-GGTTCC, the somatic cell counts were 519.03 × 10^3^ sc/mL; for TTTTTTTTAATCAATCAAGGCC, they were 1080.97 × 10^3^ sc/mL; and for TTTCTCTA.-AATCGATGCC, they were 2025.01 × 10^3^ sc/mL. Structural differences were found based on the presence of alleles C, C, A, .-, G, and T at SNPs 39, 40, 41, 42, 43, and 44.

## 4. Discussion

Grouping SNPs into haplotype blocks combines information of adjacent SNPs into composite loci haplotype alleles. Some authors suggest that assessing haplotypes may provide more robust and powerful information than considering individual SNPs, which may also translate into a higher ability to capture the regional LD information. As a result, this may enhance the understanding of genetic variability and, in turn, the combined regulation of phenotypic expression by different heritable units [43].

This has been suggested for milk quality traits, such as somatic cell counts [44], as genome-wide association studies (GWASs) using haplotypes provided some additional information that was not reported when using SNPs. However, in many cases, the haplotypes are not much more informative than a single SNP because the SNPs in high LD provide redundant information [45].

In this regard, Liu et al. [46] proposed combining SNPs into functional haplotypes based on the additive and epistatic effects among SNPs. As shown by simulation studies, the haplotype-based approach clearly outperformed the SNP-based approach unless the minor allele frequency of the SNPs making up the haplotypes is low and the linkage disequilibrium between them is high. For these reasons, by comparing the sequences of each haplotype and determining the single nucleotide variations across them, we may infer which particular changes may be responsible for the differential regulation of the expression of milk yield, milk composition, and lactation curve shape parameters. Statistically significant differences in the average quantity and composition of Murciano-Granadina goat milk across casein complex haplotypes was found. Although examples in the literature are scarce, these results agree with those for Holsteins [8] and Norwegian goats [47] and similar results were also found for somatic cell counts across casein complex haplotypes as reported by Perna et al. [48]. The relationship of these genetic units may be relevant from an economic perspective as not only milk yield and composition is affected but cheese performance may be indirectly conditioned by the haplotypic sequences present on each animal or group of animals [48].

Casein nucleotide polymorphisms’ distribution and the extent of genomic linkage disequilibrium are shaped by demographic events occurring along the history of conformation of a breed (gene flow, drift, and bottlenecks). As a result, haplotype variation estimation provides indirect evidence of the processes of selection, migration, or admixture in dairy goat populations [49]. The ratio between the number of haplotypes found in the Murciano-Granadina breed and the number of animals in our sample was 1.85 animals per haplotype. These numbers were higher than those reported for other goat breeds, such as Messinese (1.15), Argentata dell’Etna and Maltese (1.03), Capra dell’Aspromonte (1), Rossa Mediterranea (1.22), Girgentana (1.15), and Norwegian dairy goat breeds (0.42).

As suggested by Hayes et al. [47], the low number of haplotypes per locus and the existence of unique private haplotypes in the Norwegian dairy goat reflects a markedly low variability within the casein cluster, which may promote the distinction of this breed within the dairy goat panorama. For instance, the existence of a low number of founders and selection management driven through planned matings has probably reduced genetic diversity, even if the inbreeding rate, as shown by the population study through anonymous neutral marker STRs, has remained within acceptable levels [50]. Additionally, as suggested by Criscione et al. [49], the diversity of haplotypes in the Murciano-Granadina breed may be indicative of the close geographical distance to the center of goat domestication, which offers a relatively rich pool of rare specific haplotypes, which may be worth considering for conservation purposes.

The association between haplotypic sequences in the caprine chromosome 6 and milk yield, fat, and protein content has been addressed by many authors as reported by Mucha et al. [51]. For instance, caprine autosome 6 hosts a genomic window (6:86,050,148–6:86,990,478) that explains 1% of the genomic variance in milk yield [43] and comprises the *MTHFR* gene, which accounts for a known relationship with milk protein synthesis [52]. However, associations with other important economical traits, such as lactose [53] or somatic cell counts, or parameters describing the shape of the curve for either milk yield or composition are scarce in goats.

Our results suggest the GA, AA, and CC allelic combinations at the αS1-casein locus may be associated to higher milk yields (protein, fat, dry matter, and lactose), component percentages, and somatic cell counts. According to Gigli et al. [54], the presence of allele A is associated with increased milk production. Contrarily, Dagnachew et al. [35] observed that GA polymorphisms may be concomitant with lower milk yields of a higher milk yield quality in regards to its fat and protein content. Kucerova et al. [55] suggested that the presence of CC polymorphisms in haplotypic sequences may increase milk yield and protein content. This may support our results provided a statistically significant difference was found for average milk yield and protein percentage peak along lactation.

This finding may suggest polymorphisms themselves may directly affect the peaks described by protein with the independence of the variants found in the promoter region [5], which may indeed explain the difference in casein levels between a homozygote animal for highly productive alleles in comparison to one carrying double doses of a low productive allele, as suggested by Grosclaude et al. [56].

In these regards, Barbieri [57] described that goat milk presenting high levels of αS1-casein also presents a higher overall quality, which not only affects protein but also fat, total solids, phosphorus, and a lower pH compared to goat milk, which is poor in casein content. Contrastingly, the high variability found for somatic cells across casein complex haplotypes suggests the presence of enzymes, such as plasmin and proteases, from such cells may hydrolyze αS1 and β-caseins, diminishing the total concentration of proteins, especially caseins. As a result, high somatic cell counts found during the process of coagulation of milk may indirectly support the evidence of an association with casein complex haplotypes [58]. Increases in the somatic cell count in milk have been reported to negatively affect cheesemaking ability-related parameters and the sensory quality of cheese [59].

When the different haplotypic possibilities found at the αs2-casein are considered, our results suggest that a certain variability of milk yield, milk composition, and somatic cell counts may be ascribed to the different combinations of the alleles T, C, G, and A as it has already been described in the literature [5,60]. Contextually, according to Baltrėnaitė et al. [61], the sequence that presents a change towards the presence of the allele A is associated with higher milk yields.

The different haplotype combinations that can be determined when the β-casein locus is considered exert a strong favorable effect on milk yield and composition, where the presence of the alleles A, G, T, and C is related to higher production and composition percentages. Contrastingly, G and T alleles may imply a reduction in somatic cell counts. However, this contrasts the finding by Baltrėnaitė et al. [61], who did not find statistically significant differences for milk performance across the different allelic combinations within the β-casein locus. In this context, Chessa et al. [62] reported that C may be the most frequent allele to appear within the β-casein locus.

As our results suggest, when the possible haplotype combinations found at the κ-casein locus are considered, involving a group of SNPs in the promoter region, no trend for milk yield, composition, or somatic cell counts could be found. However, when β-casein haplotypes are considered together, a conjoined action seems to be exerted, which modules milk performance and composition, as it was also suggested by other authors [63].

T, C, A, and .- alleles for κ-casein and their relationship with milk yield and components are supported in the findings by Pizarro Inostroza et al. [5]. Additionally, the presence of these alleles implies lower somatic cell counts. According to Noeparvar and Morison [64], κ-casein may be related to lactose content given the role of this casein in the stabilization of calcium phosphate of milk, which is the main stabilizing agent of casein micelles [65]. Furthermore, κ-casein is the only casein that can post-translationally glycosylate via O-linked glycosylation of threonine residues [66].

Berget et al. [14] suggested that milk performance and composition may vary along the lactation curve as a result of the genetic-conditioning effects of polymorphisms occurring at the casein complex. However, Cardona et al. [67] reported that the effects of β- or κ-casein genotype possibilities may occur from half of the lactation on, which may somehow support our results for all the peaks in the curve for milk yield, fat, dry matter, lactose, and somatic cells and all persistence values not being conditioned across haplotypic sequences.

Strucken et al. [68] and Strucken et al. [15] suggested that all curve shape parameters (peak and persistence) could be considered new selection criteria, but GWAS may need to be developed. Our results complement such a statement, as only the protein percentage peak reported significant differences across the potential haplotypic sequences at the casein complex.

We must consider the fact that constant monitoring of protein in milk and, likely, of other components may render an essential practice as seeing the objective of preventing an increase in the frequencies of unfavorable effect mutations, which becomes especially important in the process of fabrication of cheese [69], as it may occur with the somatic cell count, whose genetic background has only been occasionally dealt with in goats provided it may be strongly linked to environmental conditions, pathogens’ prevalence, and the physiological status of the animal [70,71].

According to Braunschweig et al. [72], the effects of an individual locus may be misrepresented in statistical analysis, even when these are simultaneously considered in models, as their influence on milk yield and component features could be conditioned to the cumulative effect of different loci comprising the casein complex in chromosome 6. As a consequence, a better estimation of the effects of the casein complex genetic structure can be obtained when complete combinations of alleles are considered more than unique alleles, provided the close relationship that is established among the loci in the casein complex (Figure 1) [8,73,74].

None of the parameters of the curve for any of the traits considered reported a statistically significant difference across haplotypes except for the peak in protein percentage. These results suggest persistence may not depend neither on the level of production nor be conditioned by genetic structures but may be related to other parameters in the curve. This may be in line with the findings from Pereira et al. [75], who suggested curves presenting lower peaks may indeed be more persistent, which may condition the obtention of higher yields.

In the context of our results, the selection for highly productive animals may have to focus on performance itself, either its milk yield or composition, although protein peaks could also somehow be conditioned by genetic factors related to casein complex haplotypes. According to Kelm and Freeman [76], genetically selecting individuals for a greater performance may affect the proportions of components but may not affect curve shape parameters, which may support that stated other authors [77].

The implication of casein haplotypes on protein percentage peak may be linked, as Strucken et al. [15] suggested, to the higher effects of casein genes occurring during early lactation stages as a result of the higher protein content in colostrum. The same authors [78] would suggest that to understand the genetic effect of a whole lactation cycle, several genetic routes must be considered. Among these routes, the lactation curve shape may be connected or determined by genes that control the expression of feeding patterns and blood glucose levels; digestion, absorption, and transport of nutrients; secretory cells of the mammary gland; fat and adipose tissue metabolism; protein and fat synthesis, among others [79,80].

Additionally, other authors [67] have suggested that after the research performed on the genetic markers associated with milk production, a substantial number of genes whose effects are rather active during early lactation may have an intrinsic relationship with the immune response rather than on milk productive traits [15].

Even if those genes are not strictly linked to production, genes related to the immune system may condition the productivity of animals as these may determine udder health at a high-activity moment [81,82]. This suggests that monitoring the curve described by the evolution of the immune response of the udder may be a good candidate as a selection criterion, although studies deepening its genetic background must be carried out.

The inclusion of genetic factors in the genetic evaluation for economically important traits has frequently attempted seeking the maximization of the profitability of predictive models. The consideration of heritable units of a different nature and the relationships that are established among them range from the inclusion of the effect of genotypes or haplotypes to SNPs. Although predicted breeding values have been commonly reported to present the same magnitude, the consideration of genetic factors has been suggested to improve model performance through the improvement of the reliability of predicted breeding values.

For instance, as suggested by Pizarro et al. [83], the inclusion of the αS1 casein genotype as a fixed effect in mixed models to assess goat milk production and its components increases the reliability and accuracy of breeding values for some traits, such as fat and protein contents, despite slightly decreasing the accuracy of breeding values for other traits, such as milk yield and dry matter. Furthermore, heritability standard errors for the production of milk, and fat, protein, and dry matter content when the αS1 casein genotype is included as a fixed effect are lower.

The same authors reported that the accuracy and reliability for the breeding values of milk yield when the genotype of αS1 casein was included were slightly lower than those reported by the model excluding it in bucks in the range of 0.06–0.10 (RTi, R_AP_ respectively). However, for females in both models, excluding and including the genotype effect, the range was 0.04-0.07 higher for the model excluding the genotype of αS1 casein. Contrastingly, the inverse situation occurs for fat content, for which the reliability for breeding values increased in the range of 0.02–0.04 (RTi, R_AP_ respectively) for males and 0.08–0.14 for females (RTi, R_AP_ respectively) when the genotype is included as a fixed effect.

The same situation is described for the predicted breeding values of protein content as their accuracies and reliability were 0.06–0.10 (RTi, R_AP_ respectively) for the males and 0.13–0.23 (RTi, R_AP_ respectively) for females, when the genotype of the αS1 casein was included as a fixed effect in the model. On the contrary, the dry matter describes the same trend as in the production of milk, with accuracies and reliability increasing in the range of 0.11–0.14 (RTi, R_AP_ respectively) for males, and 0.18–0.21 (RTi, R_AP_ respectively) for females when the genotype of αS1 casein was not considered as a fixed effect in the model. Among the reasons stated for females presenting higher reliabilities and accuracies, Iraqi et al. [84] suggested that as the number of offspring per dam considered in the pedigree increases, the values of accuracy increase as well.

Regarding standard errors of prediction (SEP), the widest confidence ranges were reported for milk yield regardless of the sex of the animal and whether the genotype for αS1-casein was included or excluded. Despite the inclusion of the genotype not affecting the highest end of the confidence range, the lowest end was higher when genotype was included, a common trend followed by the confidence ranges of fat, protein, and dry matter components. The confidence ranges were similar, with the exception of dry matter when the αS1 casein genotype was included, which doubled the value of the confidence range when genotype was excluded. This translated into a lower risk when making decisions on animal selection for milk yield and dry matter than for the rest of the components if the αS1 casein genotype is included. Still, when genotype was considered in the model, all traits reported a high enough level of confidence that overcame the values for confidence ranges when genotype was excluded.

The same finding was reported for the correlations between the breeding values obtained for the fat and protein content (0.049–0.056), respectively. These results, together with those of the determination coefficient (R^2^), suggest the same linear relationship between the breeding values of the traits measured with independence of the inclusion of the αS1 genotype in the model.

This was also supported by Carillier-Jacquin et al. [85], who concluded that considering the αS1-casein genotype in genetic and genomic evaluations of bucks improves the accuracy of the models used and increases that range from 6% to 27%. Similarly, and although it did not reach the same increase in accuracy levels previously reported for males, in genomic evaluations carried out on female records, as in the present study, the accuracy was slightly higher (1% to 14%) than that reported by genomic models that did not involve the genotype for αS1 casein, as also suggested by the slightly higher heritability, accuracy, and reliability values of estimated breeding values.

In line with the aforementioned results, Pizarro Inostroza et al. [86] reported the lack of significant differences in the ability to predict breeding values depending on whether genetic effects derived from the study of SNP effects (either additive, dominant, or epistatic effects) were included or not. However, despite being significant, the drastic reduction in the values of reliability and accuracy of predicted breeding values when genetic factors were included suggested the fact that the inclusion of genetic effects as fixed effects may increase the estimative power and accuracy of the model used to perform genetic evaluations for economically important traits linked to milk yield and its components.

A similar finding was reported by Mucha and Wierzbicki [87] when random models considered haplotypes. These authors suggested that the inclusion of haplotypes as random effects in models may increase the precision in the estimation of breeding values of animals with unknown phenotype (whose breeding value has been inferred after the phenotypic observations from ancestors and/or descendants) while it prevents the use of high-dimensional models as compared to when models include SNPs.

Genomic selection is a useful tool to complement quantitative genetic evaluation and is currently being implemented and used more and more frequently. Its application needs cooperation between universities, biotechnology centers, researchers, producers, companies, and breeder associations to work on the development of technologies, and incorporate them into our current evaluation system in order to predict the behavior of future progeny for certain characteristics, especially those of prominent profitability.

## 5. Conclusions

The variability of the haplotypic sequences at the loci of the casein complex can be used to define selection models and strategies for economically important traits in dairy goats. However, apart from milk yield and composition traits, out of curve shape parameters, only protein percentage peaks are a potential candidate in breeding plans considering the different casein haplotypes as selection criteria. A complete definition of the haplotypes in the casein complex in goats is difficult given the high genetic variability, which is promoted when the sequences at the locus for the four caseins are considered.

## Figures and Tables

**Figure 1 animals-10-01845-f001:**
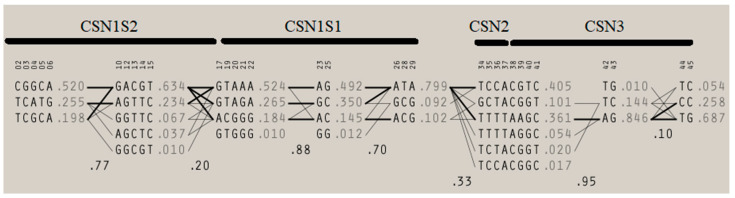
Linkage disequilibrium map across casein complex loci (CSN1S1, locus for αS1-casein; CSN1S2, locus for αS2-casein; CSN2 locus for β-casein; and CSN3, locus for κ-casein). Accessed from Pizarro Inostroza et al. [5].

**Table 1 animals-10-01845-t001:** Peak yield and persistence estimates for the lactation curve model for milk yield and composition.

Model	Parameters	Peak Yield	Persistency	Reference
Ali and Schaeffer	Milk yield (kg), protein, fat, dry matter and lactose (%)	$b_{0}$	$b_{1}\;$ $\mathrm{and}\;b_{2}$	[29]
Parabolic yield-density	Somatic cells count (SCC × 10^3^ sc/mL)	$\frac{-b_{1}}{2b_{2}}$	$2b_{0}\;Days+b_{1}$	[30]

**Table 2 animals-10-01845-t002:** Bayesian inference for one-way ANOVA to determine the differences in the mean for milk yield and components and curve parameters through best fitting models across casein haplotypes in Murciano-Granadina goats.

Trait	Parameter	Groups	Sum of Squares	df	Mean Square	F	Sig.	Bayes Factor
Milk Yield	Peak	Between	69,689.375	86.000	810.342	0.545	0.958	0.000
Within	22,301.205	15.000	1486.747			
Persitence (b1)	Between	1.822	86.000	0.021	0.608	0.921	0.000
Within	0.523	15.000	0.035			
Persitence (b2)	Between	0.000	86.000	0.000			0.000
Within	0.000	15.000	0.000			
Kg	Between	18,562.178	86.000	215.839	17.316	0.001	120,000,000
Within	37,643.494	3020.000	12.465			
Fat	Peak	Between	177,982.014	86.000	2069.558	0.818	0.729	0.000
Within	37,941.157	15.000	2529.410			
Persitence (b1)	Between	5.985	86.000	0.070	1.208	0.355	0.000
Within	0.864	15.000	0.058			
Persitence (b2)	Between	0.000	86.000	0.000	0.998	0.539	0.000
Within	0.000	15.000	0.000			
%	Between	1024.046	86.000	11.908	11.52	0.001	178,000,000
Within	3120.806	3020.000	1.033			
Protein	Peak	Between	11,785.747	86.000	137.044	3.142	0.008	0.001
Within	654.156	15.000	43.610			
Persitence (b1)	Between	0.490	86.000	0.006	1.630	0.144	0.000
Within	0.052	15.000	0.003			
Persitence (b2)	Between	0.000	86.000	0.000			0.000
Within	0.000	15.000	0.000			
%	Between	253.694	86.000	2.950	16.951	0.001	311,000,000
Within	525.565	3020.000	0.174			
Dry Matter	Peak	Between	243,045.970	86.000	2826.12	1.089	0.452	0.000
Within	38,915.155	15.000	2594.34			
Persitence (b1)	Between	8.013	86.000	0.093	1.330	0.274	0.000
Within	1.051	15.000	0.070			
Persitence (b2)	Between	0.000	86.000	0.000	0.661	0.882	0.000
Within	0.000	15.000	0.000			
%	Between	1823.530	86.000	21.204	13.804	0.001	382,000,000
Within	4638.983	3020.000	1.536			
Lactose	Peak	Between	3683.784	86.000	42.835	1.238	0.334	0.000
Within	518.983	15.000	34.599			
Persitence (b1)	Between	0.164	86.000	0.002	0.906	0.634	0.000
Within	0.032	15.000	0.002			
Persitence (b2)	Between	0.000	86.000	0.000			0.000
Within	0.000	15.000	0.000			
%	Between	90.587	86.000	1.053	14.50	0.001	78,800,000
Within	219.336	3020.000	0.073			
Somatic cell counts	Peak	Between	201,479,119.002	86.000	2,342,780.45	0.374	0.998	0.000
Within	93,871,863.089	15.000	6,258,124.21			
Persitence (b1)	Between	9,286,148,083,739	86.000	107,978,466	0.569	0.945	0.000
Within	2,847,041,628,535	15.000	189,802,775			
×10^3^ sc/mL	Between	1,735,786,503.91	86.000	20,183,564	22.388	0.001	457,000,000
Within	2,722,657,222.52	3020.000	901,542.13

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
