# Peer review of "Bayesian Analysis of the Association between Casein Complex Haplotype Variants and Milk Yield, Composition, and Curve Shape Parameters in Murciano-Granadina Goats"

_animals, 2020, doi:10.3390/ani10101845_

Round 1
Reviewer 1 Report
In my opinion the revised paper merits the final acceptance.
Author Response
Reviewer 1
Comments and Suggestions for Authors
In my opinion the revised paper merits the final acceptance.
Response: We thank the reviewer for his/her kind comments.
Reviewer 2 Report
Minor comments.-
Figure 1 has already been published in J. Dairy Sci. 2020, vol. 103, pg. 8289, by the same authors, so they must include the reference to that paper.
I do not believe the first part of the paragraph between lines 200-212 is adequate and necessary in the context of the article:“In small sample size conditions, the probability of finding significant results decreases [38]. This limitation often translate into, given power issues, an increased hardness to obtain meaningful results [39]. Given that Bayesian analyses do not assume large samples, as it would happen in maximum likelihood estimation (either it is nonparametric or parametric inference), smaller data sets can be evaluated preventing power loss and retaining precision, as suggested by Hox, et al. [40] and Lee and Song [41]. Concretely, Bayesian estimation methods have been reported to require a much smaller ratio of parameters to observations (1:3 instead of 1:5).”
Why are 3,107 milking records considered a small sample size by the authors just to justify the use of Bayesian analysis? There are frequentist statistical methods used for small sample size that do not use asymptotic properties.
On the other hand, Bayesian analysis neither assumes large samples sizes nor assumes small samples size.
So I am recommending purging those lines and rearranging the rest of the paragraph.
Main comments.-
While I accept that some drawbacks have been fixed, the problem before mentioned still remains about the difficulties to understand why in the Results section only haplotypes (….haplotypic sequence TCGCGGCCAAGACCGAGG …..) are invoked while into the Discussion section haplotypes are not mentioned at all, and SNPs alleles are mentioned without any apparent connection with the haplotypes previously defined, at least it is not possible to know the relationships between the alleles defined into the Discussion section and the haplotypes mentioned within the Results section.
Some examples from the Results section as those below:
- Lines 266-267.- “ …… significantly higher average milk yield of 3.38 kg was found for haplotypes presenting the sequence AAGGAATTAAAAGGCCAA….”
- Lines 279-280.- “…..the highest average milk yield (2.86 kg) was reported when the αs2-casein locus carried the haplotypic sequence TCGCGGCCAAGACCGAGG….”
- Lines 290-293.- “Haplotypes presenting the TCGCGGCCAAGGCCAAGG sequence at the αS2-casein locus, reported an average milk yield of 2.40 Kg and progressively decreased when the sequence changed to TTCCGATCGAGACCGACC (2.39 Kg), TTCCAATTGGGACCGGCC (1.89 Kg) or TTCCGATCGAGACCGGCC (1.47 Kg).”
But into the Discussion section, the authors do not refer to haplotypes, and mention instead , for instance, G and T alleles….. Some examples:
- Lines 431-432.- “Our results suggest GA, AA and CC allelic combinations at αS1-casein locus may be associated to higher milk yields….”
- Lines 463-464.- “…Contrastingly, G and T alleles may imply a reduction in somatic cells counts.…..”
It looks like the content of the Discussion section and the content of the Results section are disconnected. At least I cannot understand the relationship.
Author Response
Reviewer 2
Comments and Suggestions for Authors
Minor comments.-
Figure 1 has already been published in J. Dairy Sci. 2020, vol. 103, pg. 8289, by the same authors, so they must include the reference to that paper.
Response: Citation was added as suggested by the reviewer.
I do not believe the first part of the paragraph between lines 200-212 is adequate and necessary in the context of the article:“In small sample size conditions, the probability of finding significant results decreases [38]. This limitation often translate into, given power issues, an increased hardness to obtain meaningful results [39]. Given that Bayesian analyses do not assume large samples, as it would happen in maximum likelihood estimation (either it is nonparametric or parametric inference), smaller data sets can be evaluated preventing power loss and retaining precision, as suggested by Hox, et al. [40] and Lee and Song [41]. Concretely, Bayesian estimation methods have been reported to require a much smaller ratio of parameters to observations (1:3 instead of 1:5).”
Why are 3,107 milking records considered a small sample size by the authors just to justify the use of Bayesian analysis? There are frequentist statistical methods used for small sample size that do not use asymptotic properties.
On the other hand, Bayesian analysis neither assumes large samples sizes nor assumes small samples size.
So I am recommending purging those lines and rearranging the rest of the paragraph.
Response: We followed the reviewer comment and agree with his/her suggestions. We have considered the experimental units (goats) instead of the observational units (observations), hence, that is the reason of our justification of using Bayesian inference for small sample size contexts.
Main comments.-
While I accept that some drawbacks have been fixed, the problem before mentioned still remains about the difficulties to understand why in the Results section only haplotypes (….haplotypic sequence TCGCGGCCAAGACCGAGG …..) are invoked while into the Discussion section haplotypes are not mentioned at all, and SNPs alleles are mentioned without any apparent connection with the haplotypes previously defined, at least it is not possible to know the relationships between the alleles defined into the Discussion section and the haplotypes mentioned within the Results section.
Some examples from the Results section as those below:
- Lines 266-267.- “ …… significantly higher average milk yield of 3.38 kg was found for haplotypes presenting the sequence AAGGAATTAAAAGGCCAA….”
- Lines 279-280.- “…..the highest average milk yield (2.86 kg) was reported when the αs2-casein locus carried the haplotypic sequence TCGCGGCCAAGACCGAGG….”
- Lines 290-293.- “Haplotypes presenting the TCGCGGCCAAGGCCAAGG sequence at the αS2-casein locus, reported an average milk yield of 2.40 Kg and progressively decreased when the sequence changed to TTCCGATCGAGACCGACC (2.39 Kg), TTCCAATTGGGACCGGCC (1.89 Kg) or TTCCGATCGAGACCGGCC (1.47 Kg).”
But into the Discussion section, the authors do not refer to haplotypes, and mention instead , for instance, G and T alleles….. Some examples:
- Lines 431-432.- “Our results suggest GA, AA and CC allelic combinations at αS1-casein locus may be associated to higher milk yields….”
- Lines 463-464.- “…Contrastingly, G and T alleles may imply a reduction in somatic cells counts.…..”
It looks like the content of the Discussion section and the content of the Results section are disconnected. At least I cannot understand the relationship.
Response: In the context of the scarcity of studies evaluating haplotypes in goats for the traits evaluated in the present paper, the analysis of results was performed through the comparison of haplotypic sequences. Haplotypic sequences were compared to identify which sections within them were common or on the contrary which varied across haplotypic variants. As reported in Table S2, a colour code was assigned at random to identify the same haplotypic sequence across haplotypic variants. Then afterwards, significant differences in the levels of traits (milk yield and composition and their curve shape parameters) were ascribed to the differences found in the haplotypic sequences (allelic variants) across haplotypic variants. Provided the fact that the lower number of haplotypic variants was found for the αS1-casein results will be presented accordingly, comparing this to all potential haplotypic combinations found across the rest of casein complex loci.
Reviewer 3 Report
GENERAL COMMENTS
This study is interesting, and brings “relative” new information on goat breeding programs. However, serious “misunderstandings” taking into accounting the used Bayesian inference must be revised (for instance the use of P-vales, which is a completely frequentist concept). Additionally, the main question is the lack of “reliable” application of “Casein Complex Haplotype Variants” in selection strategies. It is quite important, and was not well exploited in this manuscript. In summary, how to exploit the results reported here under a “genetic selection” viewpoint? It is not so clear when thinking under a “goat breeding program” framework.
ABSTRACT
Line 32: “curve shape parameters were used using a Bayesian inference for ANOVA.”
This sentence no makes sense. The name ANOVA (Analysis of Variance) is based on F-test, which is a “frequentist method” (assuming distributions for the “parameter estimators”, and not for the “parameters”). On the other hand, under a Bayesian approach, the distributions are assumed for the “parameters”. The used definition “Bayesian inference for ANOVA” is unsuitable, since the concepts are quite different. Suggestion: “curve shape parameters were used using a Bayesian inference”. The term ANOVA must be omitted.
Lines 37-38: “Statistically significant differences (P<0.05) were found for milk yield and components”
This is a very complex issue in the present manuscript. If the authors highlighted in the tittle “Bayesian Analysis of the Association between……”, how to exploit the ‘’P-value” concept to infer on significances? This concept does not exist in Bayesian inference. P-value is an “exclusive concept” of frequentist inference. If some kind of transformation of “posterior probabilities” (Bayesian inference) to “P-values” was used here, it must be reported and highlighted in order to avoid misunderstandings.
Lines 42-43: “casein complex haplotypes can be considered in selection strategies for economically important traits in dairy goats.”
How to use this kind of information in in selection strategies? It is quite important and was not well exploited in this manuscript.
INTRODUCTION
Lines 52-53: “As a result, geneticists are able to identify and select those individuals with superior genetic potential”
This issue is highly relayed to the last one (“Lines 42-43:”). It is not clear for the readers (including the information from the full manuscript). How the applications of these “association” techniques “are able to identify and select those individuals with superior genetic potential”? It is quite clear under a “genomic prediction framework”, but completely unclear in terms of “association genetics”. I suggest elucidating this issue in order to improve the relevance of the present study in the field “goat breeding programs”.
Lines 67-69: “The approaches that are regularly followed to perform association analyses [11] normally overlook this genetic epistatic effects change during lactation [12], which renders the study of genetic association inefficient.”
Returning to the issues reported at Lines 52-53 and 67-68, it is clear that “genetic association studies” are inefficient for “goat breeding programs”. However, the contents justifying the proposed “Haplotype Variants” are not enough to provide improvements in breeding programs. I just would like to highlight that INTRODUCTION section is not so clear for breeders that have interest in adopt SNP marker information in selection programs. Please, the authors’ opinions is very important to elucidate the question involving SNP information for “association genetics” and “genomic selection”(or “genome prediction”). Information on the “last one” was not reported in this manuscript, but it is undoubtedly the “only method” successfully used in goat breeding programs.
Lines 87-88: “Revealing which genetic units may be responsible for the higher productive success though the modellization of lactation curve may translate in a higher final profitability of milk as a product.”
Firstly, the term “modellization” is not used in the field of animal breeding. I suggest changing this term for “modeling”.
Lin2s 222-227: “As suggested in public document IBM SPSS Statistics Algorithms version 25.0 by IBM Corp. [26] Bayesian inference of ANOVA is approached as a special case of the general multiple linear regression model. A full description of the algorithms used by SPSS to perform Bayesian Inference on Analysis of Variance (ANOVA) in this study can be found in the public document IBM SPSS Statistics Algorithms version 25.0 by IBM Corp. [26]. The tolerance value for the numerical methods 226 and the number of method iterations were set as a default by SPSS v25.0 [24].”
I’m sorry, but the cited references (just software manuals cited below) are not enough to ensure the reliability of a Bayesian study based on ANOVA and its respective P-values. Please, would the authors to detail how to apply a frequentist test (ANOVA, i.e. F-test) under a Bayesian framework?
- IBM Corp.. IBM SPSS Statistics for Windows, 25.0; IBM Corp: Armonk, NY, 2017”
- IBM Corp. IBM SPSS Statistics Algorithms. 25.0 ed.; IBM Corp.: Armonk, NY, USA, 2017; p 110.
Author Response
Reviewer 3
Comments and Suggestions for Authors
GENERAL COMMENTS
This study is interesting, and brings “relative” new information on goat breeding programs. However, serious “misunderstandings” taking into accounting the used Bayesian inference must be revised (for instance the use of P-vales, which is a completely frequentist concept). Additionally, the main question is the lack of “reliable” application of “Casein Complex Haplotype Variants” in selection strategies. It is quite important, and was not well exploited in this manuscript. In summary, how to exploit the results reported here under a “genetic selection” viewpoint? It is not so clear when thinking under a “goat breeding program” framework.
Response: We thank the reviewer for his/her kind comments. However, we think there have been a misunderstanding. Bayesian ANOVA is a widely used technique as suggested for the examples provided below.
- Cleophas, T.J.; Zwinderman, A.H. Bayesian Analysis of Variance (Anova). In Modern Bayesian Statistics in Clinical Research, Cleophas, T.J., Zwinderman, A.H., Eds. Springer International Publishing: Cham, 2018; 10.1007/978-3-319-92747-3_8pp. 83-89.
We agree with the reviewer on his concern about p values. However, we never stated that p values were used in Bayesian analyses. For instance, as credited in lines 225-230. The Bayes factor (BF) measures the likelihood of null and alternative hypotheses (strength of the evidence) and is used instead of p values (from frequentist approaches) to draw conclusions. Still, as suggested by Cleophas and Zwinderman [1], extrapolation between the Bayes factor used in Bayesian approaches and p values from frequentist approaches could be performed to favour the interpretability of results. The larger the BF, the more the evidence found favours the alternative hypothesis compared to the null hypothesis. We clarified this in the body text to prevent confusions.
ABSTRACT
Line 32: “curve shape parameters were used using a Bayesian inference for ANOVA.”
This sentence no makes sense. The name ANOVA (Analysis of Variance) is based on F-test, which is a “frequentist method” (assuming distributions for the “parameter estimators”, and not for the “parameters”). On the other hand, under a Bayesian approach, the distributions are assumed for the “parameters”. The used definition “Bayesian inference for ANOVA” is unsuitable, since the concepts are quite different. Suggestion: “curve shape parameters were used using a Bayesian inference”. The term ANOVA must be omitted.
Response: Bayesian ANOVA is a widely used technique as suggested for the examples provided below.
- Cleophas, T.J.; Zwinderman, A.H. Bayesian Analysis of Variance (Anova). In Modern Bayesian Statistics in Clinical Research, Cleophas, T.J., Zwinderman, A.H., Eds. Springer International Publishing: Cham, 2018; 10.1007/978-3-319-92747-3_8pp. 83-89.
- van den Bergh, D.; Van Doorn, J.; Marsman, M.; Draws, T.; Van Kesteren, E.-J.; Derks, K.; Dablander, F.; Gronau, Q.F.; Kucharský, Š.; Gupta, A.R.K.N. A Tutorial on Conducting and Interpreting a Bayesian ANOVA in JASP. LAnnee psychologique 2020, 120, 73-96.
- Ishwaran, H.; Rao, J.S.; Kogalur, U.B. BAMarray™: Java software for Bayesian analysis of variance for microarray data. BMC bioinformatics 2006, 7, 1-21.
- Geinitz, S.; Furrer, R.; Sain, S.R. Bayesian multilevel analysis of variance for relative comparison across sources of global climate model variability. International Journal of Climatology 2015, 35, 433-443.
Lines 37-38: “Statistically significant differences (P<0.05) were found for milk yield and components”
This is a very complex issue in the present manuscript. If the authors highlighted in the tittle “Bayesian Analysis of the Association between……”, how to exploit the ‘’P-value” concept to infer on significances? This concept does not exist in Bayesian inference. P-value is an “exclusive concept” of frequentist inference. If some kind of transformation of “posterior probabilities” (Bayesian inference) to “P-values” was used here, it must be reported and highlighted in order to avoid misunderstandings.
Response: We agree with the reviewer on his concern about p values. However, we never stated that p values were used in Bayesian analyses. For instance, as credited in lines 225-230. The Bayes factor (BF) measures the likelihood of null and alternative hypotheses (strength of the evidence) and is used instead of p values (from frequentist approaches) to draw conclusions. Still, as suggested by Cleophas and Zwinderman [1], extrapolation between the Bayes factor used in Bayesian approaches and p values from frequentist approaches could be performed to favour the interpretability of results. The larger the BF, the more the evidence found favours the alternative hypothesis compared to the null hypothesis. We clarified this in the body text to prevent confusions.
Lines 42-43: “casein complex haplotypes can be considered in selection strategies for economically important traits in dairy goats.”
How to use this kind of information in in selection strategies? It is quite important and was not well exploited in this manuscript.
INTRODUCTION
Lines 52-53: “As a result, geneticists are able to identify and select those individuals with superior genetic potential”
This issue is highly relayed to the last one (“Lines 42-43:”). It is not clear for the readers (including the information from the full manuscript). How the applications of these “association” techniques “are able to identify and select those individuals with superior genetic potential”? It is quite clear under a “genomic prediction framework”, but completely unclear in terms of “association genetics”. I suggest elucidating this issue in order to improve the relevance of the present study in the field “goat breeding programs”.
Lines 67-69: “The approaches that are regularly followed to perform association analyses [11] normally overlook this genetic epistatic effects change during lactation [12], which renders the study of genetic association inefficient.”
Returning to the issues reported at Lines 52-53 and 67-68, it is clear that “genetic association studies” are inefficient for “goat breeding programs”. However, the contents justifying the proposed “Haplotype Variants” are not enough to provide improvements in breeding programs. I just would like to highlight that INTRODUCTION section is not so clear for breeders that have interest in adopt SNP marker information in selection programs. Please, the authors’ opinions is very important to elucidate the question involving SNP information for “association genetics” and “genomic selection”(or “genome prediction”). Information on the “last one” was not reported in this manuscript, but it is undoubtedly the “only method” successfully used in goat breeding programs.
Response: We agree this is an important point. We expanded the discussion in this regards including the following information.
The inclusion of genetic factors in the genetic evaluation for economically important traits has been frequently attempted seeking the maximization of profitability of predictive models. The consideration of heritable units of a different nature and the relationships that are established among them range from the inclusion of the effect of genotypes or haplotypes to SNPs. Although predicted breeding values have been commonly reported to present the same magnitude, the consideration of genetic factors has been suggested to improve model performance through the improvement of the reliability of predicted breeding values.
For instance, as suggested by Pizarro, et al. [83], the inclusion of αS1 casein genotype as a fixed effect in mixed models to assess goat milk production and its components, increases the reliability and accuracy of breeding values for some traits such as fat and protein contents, despite slightly decreases the accuracy of breeding values for other traits such as milk yield and dry matter. Furthermore, heritability standard errors for the production of milk, and fat, protein and dry matter content when the αS1 casein genotype is included as a fixed effect are lower.
The same authors reported the accuracy and reliability for the breeding values of milk yield when the genotype of αS1 casein was included were slightly lower than those reported by the model excluding it in bucks in the range of 0.06–0.10 (RTi, RAP respectively). However, for females in both models, excluding and including the genotype effect, the range was 0.04-0.07 higher for the model excluding the genotype of αS1 casein. Contrastingly, the inverse situation occurs for fat content, for which the reliability for breeding values increased in the range of 0.02–0.04 (RTi, RAP respectively) for males and 0.08–0.14 for females (RTi, RAP respectively) when the genotype is included as a fixed effect
The same situation is described for the predicted breeding values of protein content as their accuracies and reliability were 0.06–0.10 (RTi, RAP respectively), for the males and 0.13–0.23 (RTi, RAP respectively) for females, when the genotype of the αS1 casein was included as fixed effects in the model. On the contrary, the dry matter describes the same trend as in the production of milk, with accuracies and reliability increasing in the range of 0.11–0.14 (RTi, RAP respectively) for males, and 0.18–0.21 (RTi, RAP respectively) for females when the genotype of αS1 casein was not considered as a fixed effect in the model. Among the reasons stated for females presenting higher reliabilities and accuracies, Iraqi, et al. [84] suggested that as the number of offspring per dam considered in the pedigree increases, the values of accuracy increase as well.
Regarding standard errors of prediction (SEP), the widest confidence ranges where reported for milk yield regardless the sex of the animal and whether the genotype for αS1-casein was included or excluded. Despite the inclusion of the genotype did not affect the highest end of the confidence range, the lowest end was higher when genotype was included, a common trend followed by the confidence ranges of fat, protein and dry matter components. The confidence ranges were similar, with the exception of dry matter when αS1 casein genotype was included, which doubled the value of the confidence range when genotype was excluded. This translated into a lower risk when making decisions on animal selection for milk yield and dry matter than for the rest of the components if αS1 casein genotype is included. Still, when genotype was considered in the model all traits reported a high enough level of confidence that overcame the values for confidence ranges when genotype was excluded.
The same finding was reported for the correlations between the breeding values obtained for the fat and protein content (0.049–0.056), respectively. These results, together with those of the determination coefficient (R2), suggest the same linear relationship between the breeding values of the traits measured with independence of the inclusion of αS1 genotype in the model.
This was also supported by Carillier-Jacquin, et al. [85] who concluded that considering αS1-casein genotype in genetic and genomic evaluations of bucks improves the accuracy of the models used in increases that range from 6 to 27%. Similarly, and although it did not reach the same increase in accuracy levels previously reported for males, in genomic evaluations carried out on female records, as in the present study, the accuracy was slightly higher (1% to 14%) than that reported by genomic models which did not involve the genotype for αS1 casein, as also suggested by the slightly higher heritability, accuracy, and reliability values of estimated breeding values.
In line with the aforementioned results, Pizarro Inostroza, et al. [86] would report the lack of significant differences in the ability to predict breeding values depending on whether genetic effects derived from the study of SNP effects (either additive, dominant, or epistatic effects) were included or not. However, despite being significant, the drastic reduction in the values of reliability and accuracy of predicted breeding values when genetic factors were included suggested the fact that the inclusion of genetic effects as fixed effects may increase the estimative power and accuracy of the model used to perform genetic evaluations for economically important traits linked to milk yield and its components.
A similar finding was reported by Mucha and Wierzbicki [87] when random models consider haplotypes. These authors would suggest the inclusion of haplotypes as random effects in models may increase precision in the estimation of breeding values of animals with unknown phenotype (whose breeding value has been inferred after the phenotypic observations from ancestors and/or descendants) while it prevents the use of high dimensional models as compared to when models include SNPs.
Lines 87-88: “Revealing which genetic units may be responsible for the higher productive success though the modellization of lactation curve may translate in a higher final profitability of milk as a product.”
Firstly, the term “modellization” is not used in the field of animal breeding. I suggest changing this term for “modeling”.
Response: The term modellization was changed to modelling across the body test as suggested by the reviewer.
Lin2s 222-227: “As suggested in public document IBM SPSS Statistics Algorithms version 25.0 by IBM Corp. [26] Bayesian inference of ANOVA is approached as a special case of the general multiple linear regression model. A full description of the algorithms used by SPSS to perform Bayesian Inference on Analysis of Variance (ANOVA) in this study can be found in the public document IBM SPSS Statistics Algorithms version 25.0 by IBM Corp. [26]. The tolerance value for the numerical methods 226 and the number of method iterations were set as a default by SPSS v25.0 [24].”
I’m sorry, but the cited references (just software manuals cited below) are not enough to ensure the reliability of a Bayesian study based on ANOVA and its respective P-values. Please, would the authors to detail how to apply a frequentist test (ANOVA, i.e. F-test) under a Bayesian framework?
- IBM Corp.. IBM SPSS Statistics for Windows, 25.0; IBM Corp: Armonk, NY, 2017”
- IBM Corp. IBM SPSS Statistics Algorithms. 25.0 ed.; IBM Corp.: Armonk, NY, USA, 2017; p 110.
Response: We added the following section to attend the reviewer suggestion.
According to Cleophas and Zwinderman [1], in traditional ANOVA, the mean values per group are squared and after adjustment for degrees of freedom divided by the squared standard deviations of the groups. The F-statistic, should be larger than approximately 5 for the null hypothesis of no difference between variability of the treatment groups to be rejected, as tested against the variability of the subjects (within the groups). Contrastingly, Bayesian ANOVA assesses the magnitude the Bayes factor (BF) as computed from the ratio of a posterior and prior likelihood distribution. The posterior is modelled from the means and variances of the measured unpaired groups, the prior is usually modelled as an uninformative prior either from the Jeffreys-Zellener-Siow (JZS) method or, equivalently, from the computation of a reference prior based on a gamma distribution with a standard error of 1. The computation of the BF requires integral calculations for accuracy purposes. But, then, it can be used as a statistical index that pretty precisely quantifies the amount of support in favour of either H1 (the difference between the unpaired means is larger than zero) or H0 (the difference between the unpaired means is not larger than zero). Among the advantages of the Bayesian approach, it provides a better underlying structure model of the H1 and H0 and maximal likelihoods of likelihood distributions are not always identical to the mean effect of traditional tests, and this may be fine, because biological likelihoods may better fit biological questions than numerical means of non-representative subgroups do.
This manuscript is a resubmission of an earlier submission. The following is a list of the peer review reports and author responses from that submission.
Round 1
Reviewer 1 Report
The Authors have investigated an interesting and novel topic and the theme has been properly described. I would like to congratulate Authors for the good-quality of this review article, the literature reported used to write theit paper, and for the clear and appropriate structure.
The manuscript is well written, presented and discussed, and understandable to a specialist readership. In general, the organization and the structure of the article are satisfactory and in agreement with the journal instructions for authors. The subject is adequate with the overall scope of Animals journal.
The work shows a conscientious study in which a very exhaustive discussion of the literature available has been carried out. The introduction provides sufficient background, and the other sections include results clearly presented and analyzed exhaustively. In order to further improve the paper's quality, I recommend to Authors to add more recently published (2018-2020) papers investigating similar topic.
So, I think that the paper merits the acceptance after very few revisions.
Author Response
Reviewer 1
The Authors have investigated an interesting and novel topic and the theme has been properly described. I would like to congratulate Authors for the good-quality of this review article, the literature reported used to write theit paper, and for the clear and appropriate structure.
The manuscript is well written, presented and discussed, and understandable to a specialist readership. In general, the organization and the structure of the article are satisfactory and in agreement with the journal instructions for authors. The subject is adequate with the overall scope of Animals journal.
The work shows a conscientious study in which a very exhaustive discussion of the literature available has been carried out. The introduction provides sufficient background, and the other sections include results clearly presented and analyzed exhaustively. In order to further improve the paper's quality, I recommend to Authors to add more recently published (2018-2020) papers investigating similar topic.
So, I think that the paper merits the acceptance after very few revisions.
Response: We thank the reviewer for his/her kind comments. Although it is difficult to find specific literature dealing with the investigation of haplotypic sequences in the chromosome 6, we included papers published from 2018 to 2020 as suggested by the reviewer.
Reviewer 2 Report
Minor comments.-
Line 49: in “….traditional mass phenotypic selection….” mass is a dispensable word
Line 50: SNP is the acronymic of Single Nucleotide Polymorphism, so Small must be replaced by Single
Line 195: The authors refer 87 different haplotypes but in Table S2 there are only 86.
The title of Figure 1 should include the meaning of CSN1S1…..because nowhere else in the text does the meaning of these acronyms appear.
The authors consider 159 goats and an average of 4 lactations per goat, so 4 x 159 = 736 lactations, why do they only use 399 lactations?
Main drawbacks.-
The available information to estimate the effect of 86 haplotypes come from around 400 lactations so, on average, there are <5 lactations to estimate the mean of each haplotype.
Why not use non-parametric statistical methods instead of Bayesian methods in which the priors can have a very high weight in the final result? It is evident that if the registered information is scarce, the a priori information will have greater relevance.
The authors estimate the parameters of the lactation function, those parameters have an error estimate associated but these errors are not taken into account when they use those parameters as dependent variables into the ANOVA to contrast between levels of haplotypes.
Although it is not enough clear, it looks like the independent variables in the ANOVA are the individual casein haplotypes, e.g., effect of the CSN1S1, CSN1S2, CSN2 and CSN3 haplotypes, with 65 levels as a whole. Apparently, the authors detect significant interactions between those effects, so it should be of interest to know what combinations of those casein- haplotype effects have a significant interaction. For instance, is there interaction between alfaS1 and kappa-caseins?
This is relevant because if an interaction between two caseins exist what happen with the effect of the individual caseins is not relevant.
It is not very well understood, it is not very clear, and I consider must be clarify, how references to alleles defined by a single nucleotide (A, G, T, C) in the discussion section (eg lines 391, 396, 397, 406 ... ) are linked with what is expressed in the results section in which there are no references to nucleotides but to haplotypes.
Author Response
Reviewer 2
Minor comments.-
Line 49: in “….traditional mass phenotypic selection….” mass is a dispensable word
Response: The word mass was removed.
Line 50: SNP is the acronymic of Single Nucleotide Polymorphism, so Small must be replaced by Single
Response: Replaced.
Line 195: The authors refer 87 different haplotypes but in Table S2 there are only 86.
- Response: There was a tipo. We corrected it. There were 86 haplotypes.
The title of Figure 1 should include the meaning of CSN1S1…..because nowhere else in the text does the meaning of these acronyms appear.
Response: Added.
The authors consider 159 goats and an average of 4 lactations per goat, so 4 x 159 = 736 lactations, why do they only use 399 lactations?
Response: Lactation number did not normally distribute accross individuals, hence calculations made by the reviewer may not be aplicable. We added dispersion measurements (average of 3.91±2.01 lactations/goat)
Main drawbacks.-
The available information to estimate the effect of 86 haplotypes come from around 400 lactations so, on average, there are <5 lactations to estimate the mean of each haplotype.
Why not use non-parametric statistical methods instead of Bayesian methods in which the priors can have a very high weight in the final result? It is evident that if the registered information is scarce, the a priori information will have greater relevance.
Response: In small sample size conditions, the probability of finding significant results decreases [1]. This limitation often translate into, given power issues, an increased hardness to obtain meaningful results [2]. Given that Bayesian analyses do not assume large samples, as it would happen in maximum likelihood estimation (either it is nonparametric or parametric inference), smaller data sets can be evaluated preventing power loss and retaining precision, as suggested by Hox, et al. [3] and Lee and Song [4]. Concretely, Bayesian estimation methods have been reported to require a much smaller ratio of parameters to observations (1:3 instead of 1:5). Furthermore, although the simplest and perhaps the most frequently used test for parametric associations considers the relationship between a single marker and a quantitative trait, the power of this method may suffer because a single SNP may have only low linkage disequilibrium (LD) with the causal mutation and the LD contained jointly in flanking markers is ignored. Alternatively, fitting hereditary units such as SNPs or haplotypes simultaneously, using Bayesian methods, thus considering the LD between neighboring SNPs, may prevent false positive from occurring [5].
The authors estimate the parameters of the lactation function, those parameters have an error estimate associated but these errors are not taken into account when they use those parameters as dependent variables into the ANOVA to contrast between levels of haplotypes.
Although it is not enough clear, it looks like the independent variables in the ANOVA are the individual casein haplotypes, e.g., effect of the CSN1S1, CSN1S2, CSN2 and CSN3 haplotypes, with 65 levels as a whole. Apparently, the authors detect significant interactions between those effects, so it should be of interest to know what combinations of those casein- haplotype effects have a significant interaction. For instance, is there interaction between alfaS1 and kappa-caseins? This is relevant because if an interaction between two caseins exist what happen with the effect of the individual caseins is not relevant.
Response: A complete analysis of the interaction among the genes in the casein complex was performed in a paper published earlier this year using the same data sample. For instance, in this paper it was stated that the recombination access points found in the CSN3 (kappa-casein) locus may be the basis for the lack of intergroup explanatory potential of the variance of κ-casein, which may suggest a lack of interaction between this gene and the rest of the genes comprising the casein complex. We clarified haplotypic variants were considered the independent factor in the study while milk yield and composition traits were the dependent variables.
- Pizarro Inostroza, M.G.; Landi, V.; Navas González, F.J.; León Jurado, J.M.; Martínez Martínez, M.d.A.; Fernández Álvarez, J.; Delgado Bermejo, J.V. Non-parametric analysis of casein complex genes epistasis and their effect on phenotypic expression of milk yield and composition in Murciano-Granadina goats. Journal of Dairy Science 2020, 103, 8274-8291.
No interaction between haplotypic variants was considered as suggested by papers studying independent factors and the effects of their double and triple interactions. For instance, the consideration of factors independently, may help quantify their effects more accurately than their conjoint effects [6]. This may be ascribed to the fact that when a limited sample is available, some of the potential combinations across categorical factors may be misrepresented [7].
It is not very well understood, it is not very clear, and I consider must be clarify, how references to alleles defined by a single nucleotide (A, G, T, C) in the discussion section (eg lines 391, 396, 397, 406 ... ) are linked with what is expressed in the results section in which there are no references to nucleotides but to haplotypes.
Response: We clarified this in the body text. We discussed the results obtained in the light of and to fill the gap of literature information (papers dealing with SNP association within the casein complex are less scarce than those dealing with haplotype association in the same complex) and given the fact that grouping SNPs into haplotype blocks combines information of adjacent SNPs into composite loci haplotype alleles. Some authors suggest that assessing haplotypes may provide more robust and powerful information than considering individual SNPs, which may also translate into a higher ability to capture the regional LD information. As a result, this may enhance the understanding of genetic variability and in turn, the combined regulation of phenotypic expression by different heritable units [8].
These have been suggested for milk quality traits such as somatic cell counts [9], as GWAS using haplotype provided some additional information that was not reported when using SNP. However, in many cases the haplotypes are not much more informative than a single SNP because the SNPs in high LD provide redundant information [10].
In these regards, Liu, et al. [11] proposed combining SNPs into functional haplotypes based on the additive and epistatic effects among SNPs. As shown by simulation studies, the haplotype-based approach clearly outperformed the SNP-based approach unless the minor allele frequency of the SNPs making up the haplotypes is low and the linkage disequilibrium between them is high. For these reasons, comparing the sequences of each haplotype and determining the single nucleotide variations across them, we may infer which particular changes may be responsible for the differential regulation of the expression of milk yield, milk composition and lactation curve shape parameters.